# Cloud Database Tuning with Reinforcement Learning

**Chenxia Han**
Department of Computer Science and Engineering
The Chinese University of Hong Kong
Shatin, Hong Kong
cxhan@cse.cuhk.edu.hk

**Chaokun Chang**
Department of Computer Science and Engineering
The Chinese University of Hong Kong
Shatin, Hong Kong
ckchang@cse.cuhk.edu.hk

## Abstract

In database management systems (DBMSs), especially cloud DBMSs, configuration tuning is one of the key factors that influence database performance. For a long period of time, the tuning job of databases is done by experienced database administrators (DBAs), which is time-consuming and sub-optimal. Recently, with the development of machine learning, automatic tuning tool starts to play a significant role in DBMSs. Among all the learning-based methods, reinforcement learning has the greatest potentiality to find the optimal or near-optimal configuration. There are already some reinforcement learning applications in database tuning, nonetheless, none of them provide fully executable code. In this project, we implement an independent version of auto-tuner to re-produce the current works on MySQL. Based on a careful design of the system, our code can be easily applied to other databases like JDB, LevelDB, etc. We believe it will benefit a lot to those database administrators who know a lot about databases but lack the knowledge of reinforcement learning. The code is available at `https://github.com/ChaokunChang/GDBTuner`. The 5 minutes video can be found at Google Drive[1].

## 1 Introduction

The configuration file of a database is usually composed of numerous tunable parameters, which can be represented by key-value pair and usually being called as knobs. The performance of databases are highly influenced by the value of these knobs. With a proper set of knobs, the DBMSs will have higher throughput and lower latency, and vice versa. Therefore, knob tuning becomes an important part of DBMSs. However, the tuning problem of DBMSs is NP-hard, it's impossible to enumerate all the possible values of hundreds of knobs. Hence, only a small number of database administrators (DBAs) have the ability to tune the knobs with heuristic rules and their experience. There are three problems to tune the configuration with this manual method: 1) It's expensive to train a DBA to be professional. 2) DBAs rely on some heuristic rules and experience, which may not be effective when adding new knobs. 3) It is infeasible to let DBAs tune every database instance in a cluster with hundreds or thousands of machines. As a result, automatic tuning plays an important role in DBMSs.

This problem is also interested by researchers and engineers in the industry. In a production environment, a common database business is supported by hundreds of servers, and each server may have

---

[1]`https://drive.google.com/drive/folders/1Z1MvtBnQ522zPElNu-VSrAMvKcX523LE?usp=sharing`

several database instances. Different database instances are deployed in different environments, like different data, different workload, different hardware, etc. In this case, it will hurt the performance if all database instances share the same configuration file. At the same time, it is not plausible to tune the database instances using time-consuming methods like DBA manually tuning. Thus, machine learning and RL techniques can help engineers to tune database cluster easily, only with few human efforts.

The techniques for automatic database tuning can be divided into two classes. The first one is rule-based search methods, e.g. BestConfig(14), which utilize heuristic method to search the optimal solution according to history. The other one is learning-based methods, e.g. OtterTune(12), CDBTune(13). OtterTune can tune the database by learning the experience from the historical data, thus the performance of OtterTune highly depends on the number and quality of training data, which is hard to obtain. CDBTune, in contrast, replaces traditional machine learning with reinforcement learning, which tuning the database with a try-and-error strategy. Through the experiments of CDB-Tune, reinforcement learning outperforms other learning techniques like linear regression, CNN, etc. However, the author of CDBTune didn't fully release their source code.

In this paper, we present our design and implementation that can automatically tune the database instance using reinforcement learning.

## 2 Related Work

Many researchers have tried different techniques to help DBA to tune the knobs and evaluate the expected performance after tuning.

Peloton (5) is the first self-driving DBMS, which is capable to forecast workload and deploy action. It first classifies the workloads to several groups to reduce the number of forecast models. To forecast the workload of each group, it tags each query and populates a histogram for each cluster within a period of time. *Recurrent neural networks* (RNNs) is adopted to predict this kind of time-series patterns. Then, Peloton searches for the best action in history so that it benefits most in the forecasting environment.

Similarly, OtterTune (11; 12) uses both supervised and unsupervised learning to enable automated knob tuning. It clusters internal runtime metrics via k-means and selects one representative metric closest to the center of the cluster to remove redundant metrics. In the configuration recommendation phase, it finds the workload in history with the most similarity and fits the Gaussian Process Regression model to recommend knobs.

At the same time, some researchers want to save more resources of DMBSs with the least sacrifice of performance. iBTune (10) utilizes the relationship between cache miss ratios and memory size to estimate the best buffer size by large deviation analysis. And to guarantee service level agreement (SLA), they build a DNN model to predict response time so that configuration with unqualified expected SLA would not be considered.

Recently, Zhang (13) proposes the first end-to-end automatic database tuning system CDBTune, with the power of reinforcement learning (RL). Due to the advantage of RL, it completely alleviates the difficulty of collecting data for model training. Compared with previous traditional regression methods, a reward-feedback mechanism is born to fit the knobs tuning problem. Just as how DBAs learn to tune the database, the agent makes the decision of knobs change and deploy it to the database. After a period of time, the agent can observe performance change and learns knowledge from this action.

CDBTune shares the most similar behavior with experienced DBA, which adopts trial-and-error method with the help of RL. They use DDPG as their agent to predict continuous action. However, DDPG is only one of the value-based family.

There are many algorithms in reinforcement learning, but all of them can be divided into two types: value-based and policy-based RL. Valued-based RL tries to solve the problem through dynamic programming. It evaluates the state value function and state-action value function. The representative algorithms are Deep Q-learning and its variant DDPG (9), which can handle continuous space action. By contrast, policy-based RL doesn't obtain the policy from the value function but directly learns a policy function that takes the state as input and outputs the probability to take the next action. Since

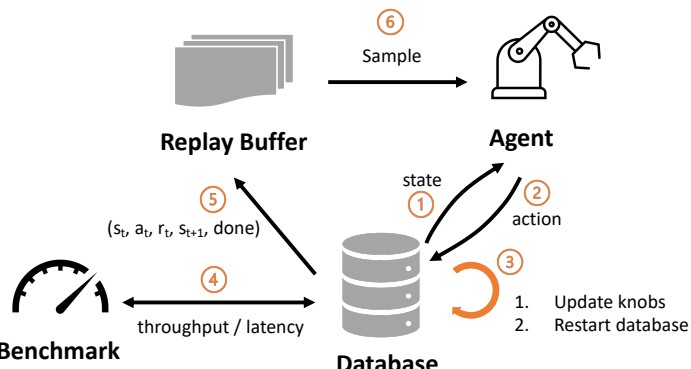

Figure 1: System Architecture

policy-based RL learns the probability distribution between state and action, it has a natural advantage to learn uncertain policy, and also compatible with the action of continuous space. The representative works of policy-based RL are Policy Gradient and its variants such as TRPO (7) and PPO (8).

## 3 Framework

In this section, we describe the key components in the RL framework, including state, action space, and reward function. The agent in our method is the tuning system which receives reward and state from DBMSs and then recommends knobs. And the environment is the database instance itself, which show different performance when changing configurations.

Figure 1 shows the architecture of our system. Next, we elaborate on the meaning of each component:

**State:** Database command "show status" can provide internal metrics as the current state of the RL agent. Those metrics, such as how many pages read/write during the period around time t, are vectorized as $s_t$.

**Action:** There are many tunable knobs in the database configuration, like `table_open_cache`, `innodb_buffer_pool_size` and `innodb_buffer_pool_instances`. Besides, constraint of hardware resources should also be considered, such as the number of hyper-threading and DRAM usage. Here we define the value of tuning knobs as the action $a_t$.

**Reward:** Reward is designed to penalize the agent if the performance drops when applying this action. Thus, we take the performance change before and after applying new knobs as reward $r_t$.

As our action space is continuous and very large, we adopt Deep Deterministic Policy Gradient (DDPG) (9) as our RL algorithm to tackle this issue. DDPG takes advantage of both DQN (2) and Actor-Critic algorithm and can learn the policy from high-dimensional actions.

Unlike DQN, DDPG learns a deterministic policy $\mu_\theta(s)$ to directly give the action the maximizes $Q_\phi(s, a)$, i.e. $\arg\max_a Q^*(s, a) \approx Q_\phi(s, \mu_\theta(s))$. The formula of Q-function is given as following:

$$\min \quad E_{s,r,s',d \sim \mathcal{D}}[Q_\phi(s, a) - y(r, s', d)]$$
$$\text{subject to} \quad y(r, s', d) = r + \gamma(1 - d)Q_{\phi_{targ}}(s', \mu_{\theta_{targ}}(s'))$$

where $y$ is the Q target, $r$ is the reward and $\gamma$ is the discount factor. Here $d$ is either 1 or 0, indicating current episode is done or not. And the objective function is:

$$\max_\theta E_{s \sim \mathcal{D}}[Q_\phi(s, \mu_\theta(s)] \tag{1}$$

After solving the above problem, we can obtain the policy $\pi_\theta$, which maximizes the expectation of rewards.

As the goal of Policy Gradient is to maximize the expected reward, we design the reward function as following. There are two major metrics to evaluate database performance: throughput and latency of responding to queries. Whenever the agent tunes parameters of database, and the performance becomes better, the reward of this action should be positive. While if it's worse, the reward should be negative. Thus, we take the performance change during each episode:

$$\Delta T_{i \to j} = \frac{T_j - T_i}{T_i} \tag{2}$$

$$\Delta L_{i \to j} = \frac{-L_j + L_i}{L_i} \tag{3}$$

The above equations mean, in timestamp $i$ and $j$, its throughput is $T_i$ and $T_j$ respectively, then $\Delta T_{i \to j}$ is the normalized performance change. As for latency, since the smaller one is better, its sign is opposite with throughput in Equation 3. Since we want the ultimate performance of database is better than initial setting, we should take the performance change between current and the initial one into consideration. And it should have more impact than intermediate performance change. To reduce the impact of intermediate setting, we design the reward as following:

$$r = \begin{cases} ((1 + \Delta_{0 \to t})^2 - 1)|1 + \Delta_{t^* \to t}|, & \Delta_{0 \to t} > 0 \\ -((1 - \Delta_{0 \to t})^2 - 1)|1 - \Delta_{t^* \to t}|, & \Delta_{0 \to t} \leq 0 \end{cases} \tag{4}$$

where $t^*$ indicates the timestamp with the best performance in each episode. It aims to push the agent to perform better than the previous best action in the episode. To combine the two metrics, we denote the weighted sum of throughput and latency reward:

$$R = c_T * r_T + c_L * r_L \tag{5}$$

## 4 Implementation Details

In this section, we introduce the implementation details of our system for database tuning, including the environment design, model design, and training progress. To make our system more scalable, we adopt Ray (3) framework to train our agent. Since Ray provides implementations of DDPG and Prioritized Replay Buffer (6), we only need to wrap our environment into Gym (1) style. What's more, to make sure our work is easy to reproduce and generalize, we carefully designed our system. Researchers only need to simply write a database connector to apply our system in their own database.

### 4.1 Environment Implementation

To apply reinforcement learning methods in the database tuning problem, the first step is to design the environment which can represent database tuning. We follow the format of Gym to implement our own environment, in which only three interfaces (1) init (2) reset (3) step are exposed. We introduce them separately.

**init:** The most important thing we do in this progress is to create a database handle that holds the information of the database and its connection, and a simulator handle that holds the commands to execute workload simulation test.

**reset:** This function is called at the beginning of each episode to reset the environment, including the knobs, state metrics, and simulation performance.

**step:** This is the most important function which is called at each steps. The progress of this function is showed in Figure 2. We first generate knobs according to the action given by RL model, and then apply the knobs to the database. After that, we get the states of current database. Finally, we compute the reward for current state and action.

Next, we elaborate on the details in the above three functions.

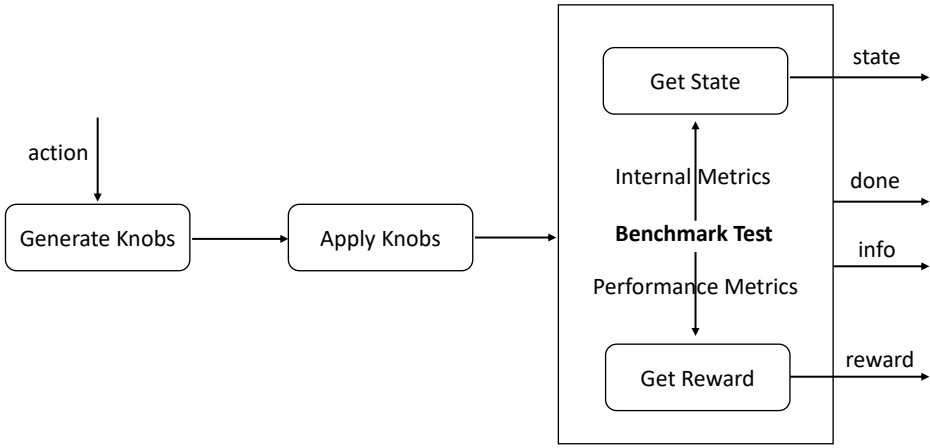

Figure 2: Detail of step function in environment

**Action to Knobs:** The action given by the RL model is an array of continuous value between 0 and 1. We map these values to specific knobs values according to their knob types and value intervals.

Though there are more than 400 metrics in the query result of "SHOW STATUS", we only take the enabled 74 metrics as state. Besides, we manually pick 6 metrics as action space, i.e. only 6 knobs are controlled in each step, according to the importance ranking provided by Ottertune. Each time the agent predicts a 6-dimensional tensor and each dimension is bounded to $[0, 1]$. Then the knobs in next step are derived by following:

$$\text{Knob}_{i+1,j} = v_{\min}^j + \lfloor (v_{\max}^j - v_{\min}^j) * a_i^j \rfloor \qquad \text{for } j = 0, 1, \cdots, 5$$

where $v_{\min}^j$ and $v_{\max}^j$ is the minimum and maximum value of $j$-th knob respectively. And $\text{Knob}_{i+1,j}$ refers to the $j$-th entry of knobs in $(i + 1)$-th timestamp. Since all of these 6 knobs are integers, we take floor operation in the above equation. Note that our action doesn't predict the change of knobs, it predicts the value of knobs directly. We also extend our knob set to a larger one, with 60 tunable knobs and supports various types, including **integer, float, enum and boolean**. The calculation from action to knobs are similar for other types, thus we omit the details here.

**Apply Knobs:** To apply knobs to the MySQL database, we first write the knobs to the configuration file of database and then restart the database to activate the new knobs.

**Generate State:** For the representation of the environment state, we borrow the idea from DBA and other auto-tuning systems. That is, using the internal metrics to represent the current workload, i.e. the state of the environment in the RL system. We use 74 typical internal metrics to represent our environment, which is enough for our setting. These metrics can be divided into two types: Value type and Counter type. For the value type, we calculate the average value during a period of time (75s in our experiments); For the counter type, which continuously increases during the running of the database, we calculate the increased number during the period.

As we do not have real-world workload, we use SysBench to generate a simulating workload for training and report the current throughput and latency as external metrics to evaluate how good the current state and action are. What must be mentioned is, the progress of the SysBench test and the progress of state generation are executed asynchronously. We run SysBench for 75s in each step, and collect the internal metrics and external metrics at the same time.

**Calculate Reward:** To calculate the reward of the current step, we use the external metrics collected from the former stage. The algorithm to calculate reward is already introduced in the above sections. To be more explicit, we multiply the positive reward with a big number to enlarge the influence of positive reward, as it's hard to get a good combination of knobs in this environment.

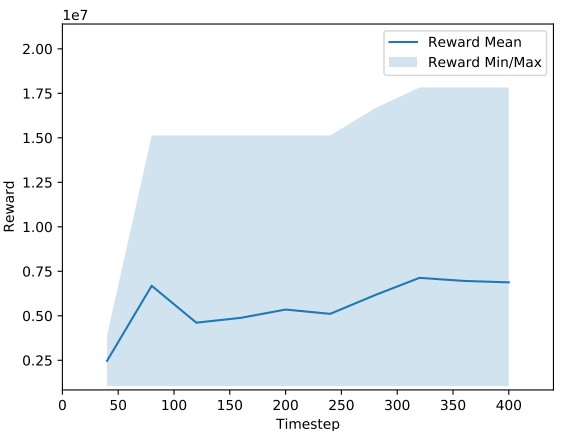

Figure 3: Timestep vs. Reward

## 4.2 Model Design

We build an Actor and Critic network following the default network in Ray. The Actor network has two fully connected layers, each has $400$ and $300$ hidden layers, followed by ReLU and BatchNorm layers. The Critic network has exactly the same fully connected architecture as the Actor.

## 5 Experiments

To estimate our auto tuner, we did some experiments on MySQL installed on a single Linux server. We choose 6 knobs instead of all the hundreds tunable knobs for our experiments because the training of our agent is extremely time-consuming and it shows in CDBTune and OtterTune that these 6 metrics have the biggest impact on the performance.

In the experiments, we set the batch size as $16$ and learning rate as $10^{-5}$ for both Actor and Critic networks. The discount factor $\gamma$ is set to $0.99$ and $\tau$ is set to $10^{-5}$. Also, the weight of reward $c_T$ and $c_L$ are set to $0.4$ and $0.6$ respectively to balance the penalty of throughput and latency.

Different from most game environments, ours is quite slow to finish one step, i.e. several minutes, as the benchmark tool requires more time to estimate the performance of the current database instance. Hence, we train our agent for $400$ steps currently, which is around 10 hours. But the agent can still learn experience from these few samples. Figure 3 shows the reward during the training phase, which increases over time. Due to the advantage of the off-policy algorithm DDPG, it's possible to save replay buffer into files and load them to conduct offline training to reduce training time. Also with the good scalability of Ray, it's easy to implement distributed learning where each remote server owns one database instance and serves as an environment separately to improve sampling efficiency. We leave the distributed training part as future work.

Imagine the process of searching best knobs as walking in a high-dimensional space where each coordinate corresponds to different performance. Obviously, the best place an agent ever visited is the coordinate with the best performance in one episode. Hence in our evaluation phase, we load the trained agent and run it for one episode, i.e. stop until the environment is done, and find the best knob over this episode. To show the effectiveness of our agent, we compare the performance (TPS[2]/Latency/QPS[3]) of default knobs with the best knobs the agent found in Table 1, where the default knobs are from the official documentation of MySQL (4). It is worth to mention that the agent improves over $20\%$ in both throughput and latency than default knobs. Also, a very interesting fact is that, in the evaluation, the agent only takes 5 steps to finish the episode and find the best in the 2nd step, indicating the efficiency of our method. Moreover, even without the knowledge from human but

---

[2]TPS: Transaction per second
[3]QPS: Query per second

| Knobs | TPS | Latency(ms) | QPS |
|---|---|---|---|
| Default | 3417 | 5.652 | 54676 |
| Min | 1385 | 16.296 | 22154 |
| Max | 3923 | 5.868 | 62763 |
| Ours | 4179( +**22.2**%) | 4.058(+**28.2**%) | 66869(+**22.3**%) |

Table 1: Comparison of performance between default and the best found by our agent

| Knobs | Min | Max | Default | Ours |
|---|---|---|---|---|
| table_open_cache | 1 | 524288 | 4000 | 524288 |
| innodb_buffer_pool_size | 5242880 | $2^{32}$ | 134217728 | 4294967296 |
| innodb_buffer_pool_instances | 1 | 64 | 1 | 1 |
| innodb_purge_threads | 1 | 32 | 4 | 32 |
| innodb_read_io_threads | 1 | 64 | 4 | 1 |
| innodb_write_io_threads | 1 | 64 | 4 | 1 |

Table 2: Comparison of knobs between default and the best found by our agent

only with reward from the environment, **our agent learns to increase buffer size and the number of purge threads**, as shown in Table 2.

Note that **setting all knobs as the maximum value does not lead to the best performance**, as some knobs are competitive, such as read threads and write threads. Since the CPU resource is limited and when read operation occupies most computational resources, it slows down the operation of write. Also increasing buffer size without limit degrades the performance of databases. We shows the performance of MySQL instance with all minimal and maximal knobs respectively in Table 1, which is worse than the knob found by our agent.

## 6 Conclusion

In this paper, we present our design and implementation details of database tuning solution in the reinforcement learning approach. The experiment shows the effectiveness of our method and the promising improvement on both throughput and latency in SysBench. This system can help to tune database easily, even without any domain knowledge in database tuning, and achieves comparable performance with database experts. Moreover, as our system is implemented in Ray, it's easy to scale to a large cluster to tackle the sampling efficiency issue.

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
