# OpenReview forum: "Cloud Database Tuning with Reinforcement Learning"
_CUHK.edu.hk/2021/Course/IERG5350_

### Official Review · AnonReviewer3 · 2020-12-18
**Database tuning with Rl**

**Rating:** 7
**Confidence:** 4

**Review:**

Overview:
	The paper implement a indepent version of auto-tuner to re-produce the current works on MySQL. In detail, the paper define a new environment of databse tuning used in RL and use Ray framework to train the agent.By using DDPG implemented in Ray, it gets the result that improve over about 20% in both throughput and latency than default configuration.

Overall Evaluation:
The idea has practical signifancance.
The logic of this paper is fluent.
The experimental settings are reasonable.
The results are clear.

Significance:
The paper trained a more efficient agent by DDPG. In conclusion, the configuration it gets performed better than default configuration.

Originality:
The overall framework design utilize the benchmark to evaluate rewards and the replay buffer to improve the effectiveness of episodes.
The reward function propsed in paper is novel. I think it improves the effect of training and can explore more based on the best results in an episode.

Comments:
In abstract, authors were planning to generalize the trained agent to other database. However, I don't see any content about it in the body of this paper.
Due to the limitation of training time, there may be more space for improvement in the exploration of auto database tuning.
In section3, I think it is better to explian more about the formula of Q-function.The meaning of 'd' is not clear here. Besides, I don't undertand why authors choose the 6 metrics as action space.

---

### Official Review · AnonReviewer1 · 2020-12-18
**This paper studies RL algorithms for Cloud Database Tuning**

**Rating:** 7
**Confidence:** 4

**Review:**

The report focuses on the design and implementation of reinforcement learning with DDPG algorithm to automatically adjust database instances. The idea and realization are in line with the requirements of this course.

Quality: met the acceptable quality bar as an experimentation project.
Clarity: The project framework is detailed and clear. There are grammatical errors such as singular and plural verbs and nouns.The format of the authors' information is different from the template, and perhaps the merge of the mailboxes of the two will bring unnecessary trouble.
Originality: paper is that CDBTune replaces traditional machine learning with reinforcement learning to tune the database with try-and-error strategy and rewards good jobs.
Significance: A more meaningful research direction to help engineers easily adjust database clusters without spending too much human resources. In the experiment results, it can be concluded that the agent increases over 20% in both latency and throughput latency than default knobs.